# Assessing vehicle fuel efficiency using a dense network of CO$_2$ observations

Helen L. Fitzmaurice[1], Alexander J. Turner[2], Jinsol Kim[1], Katherine Chan[3], Erin R. Delaria[4], Catherine Newman[4], Paul Wooldridge[4], Ronald C. Cohen[1,4]

1. Department of Earth and Planetary Science, University of California Berkeley, Berkeley, CA, 94720, United States

2. Department of Atmospheric Sciences, University of Washington, Seattle, WA, 98195, United States

3. Sacramento Metro Air Quality Management District, Sacramento, CA, 95814, United States

4. Department of Chemistry, University of California Berkeley, Berkeley, CA, 94720, United States

*Correspondence to*: Ronald C. Cohen (rccohen@berkeley.edu)

**Abstract.** Transportation represents the largest sector of anthropogenic CO$_2$ emissions in urban areas in the United States. Timely reductions in urban transportation emissions are critical to reaching climate goals set by international treaties, national policies, and local governments. Transportation emissions also remain one of the largest contributors to both poor air quality (AQ) and to inequities in AQ exposure. As municipal and regional governments create policy targeted at reducing transportation emissions, the ability to evaluate the efficacy of such emission reduction strategies at the spatial and temporal scales of neighborhoods is increasingly important. However, the current state of the art in emissions monitoring does not provide the temporal, sectoral, or spatial resolution necessary to track changes in emissions and provide feedback on the efficacy of such policies at a neighborhood scale. The BErkeley Air Quality and CO$_2$ Network (BEACO$_2$N) has previously been shown to provide constraints on emissions from the vehicle sector in aggregate over a ~1300 km$^2$ multi-city spatial domain. Here, we focus on a 5 km, high volume, stretch of highway in the SF Bay area. We show that inversion of the BEACO$_2$N measurements can be used to understand two factors that affect fuel efficiency: vehicle speed and fleet composition. The CO$_2$ emission rate of the average vehicle (g/vkm) are shown to vary by as much as 27% at different times of a typical weekday because of changes in these two factors. The BEACO$_2$N-derived emissions estimates are consistent to within ~3% of estimates derived from publicly available measures of vehicle type, number, and speed, providing direct observational support for the accuracy of the Emissions FACtor model (EMFAC) of vehicle fuel efficiency.

## 1 Introduction

Urban emissions currently account for ~75 % of all anthropogenic CO$_2$ emissions (IPCC, 2014). By 2050, roughly two-thirds of the earth's projected population of 9.3 billion is expected to reside within urban areas (IPCC, 2014), meaning that effective greenhouse gas emissions reductions strategies must focus on urban emissions reductions.

The transportation sector is responsible for ~23% of global greenhouse gas emissions worldwide (IPCC, 2014) and represents the greatest sectoral percentage (~25-66%) of emissions from within the boundaries of urban areas in the United States (Daw, 2020; Gurney et al., 2021). Although fuel efficiency of new internal combustion engine vehicles has increased by ~30% over the last 20 years and electric vehicles (EV) are becoming more prevalent  (e.g. https://arb.ca.gov/emfac/emissions-inventory), emissions reductions resulting from fuel efficiency gains in newer vehicles

are negated by an increasing percentage of heavy-duty vehicles (HDV) (Moua, 2020), speed-related reductions in fuel efficiency resulting from increases in congestion, and an increase of total vehicle kilometers traveled (vkm). Over the past 20 years, even in locations with aggressive climate change policy, these factors have resulted in $CO_2$ emissions from vehicles that have increased or stayed nearly constant. For example, California Air Resources Board estimates that in the state of California, per capita vehicle emissions in 2015 were only 2% lower than in 2000 and per capita vehicle kilometers traveled

(vkm) increased ~2.5% over that time period (California Air Resources Board, 2018). In addition to GHG emissions, the transportation sector is responsible for a significant share of $PM_{2.5}$ and $NO_x$ emissions, exacerbating $PM_{2.5}$ and ozone exposure in low-income communities and communities of color already experiencing disproportionate health burdens associated with poor air quality (Tessum et al., 2021).

        Municipal and regional governments have increasingly shown interest in tracking and reducing $CO_2$ emissions from

all sectors, including transportation. For example, Boswell et al. (2019) found that 64% of Californians live in a city with a climate action plan. For urban and regional governments to plan, monitor, and responsively adjust emissions reduction policies, an up-to-date understanding of the spatial and temporal variations in total emissions and in emissions by sector and subsector processes is key.

        For transportation, reductions in vkm, congestion mitigation, and rules affecting fleet composition (e.g., limiting

road access to HDV, incentivizing use of electric vehicles, or buy-backs of older vehicles) are three levers that can be employed to reduce $CO_2$ and AQ emissions from vehicles, thereby affecting the climate footprint, air quality (AQ), and environmental justice (EJ) in a region. However, the current state of the art in emissions monitoring and modelling do not provide the temporal, sectoral, or spatial resolution necessary to track changes in urban emissions and provide feedback on the efficacy of each lever separately. Furthermore, current estimates of the magnitude and sectoral apportionment, of urban

$CO_2$ emissions can vary widely. For example, Gurney et al. (2021) show how a consistent approach to total emissions from cities across the U.S. differs from locally constructed inventories in magnitude and sector by sector.

        Spatial and temporal process-level maps of emissions are needed to improve the scientific basis for emission control strategies. The current state of the art involves finding aggregate emissions over large regions (counties, states) using economic data and downscaling those totals using proxies such as road length, building type or population density. These

models meet the need for high spatial resolution (~500 m) and capture emissions from many detailed subsectors (Gately et al., 2015; Gurney et al., 2012; McDonald et al., 2014). Because fuel sales are well-characterized, these models are also likely to produce accurate region-wide $CO_2$ emissions totals from the transportation sector.

Yet even the most detailed of these inventories do not presently describe the temporal variability in processes that affect emissions, such as the direct response of home heating or air conditioning to ambient temperature or, with one exception (Gately et al., 2017b), the variations in emissions per km when comparing free-flowing to stop-and-go traffic. These models often disagree with one another spatially (Gately et al., 2017a), have been subject to only limited testing against observations of the atmosphere, and are not designed to be consistent with separately constructed AQ inventories that have been subject to much more extensive testing against observations.

Mobile monitoring campaigns and high-density measurement networks highlight the importance of characterizing and identifying the processes contributing to sharp neighborhood-scale AQ and GHG hotspots and point to the importance of traffic emissions on neighborhood scales. For example, Apte et al. (2017), showed that concentrations of $NO_x$ and Black Carbon (BC) can vary by as much as a factor of ~8 on the scale of 10s to 100s of meters. Caubel et al. (2019), showed BC concentrations to be ~2.5 times higher on trucking routes than on neighboring streets. Such gradients are not represented in inventories based on downscaled economic data.

Observations of $CO_2$ and other greenhouse gases can play an important role in improving and maintaining the accuracy of emission models—especially during a time of rapid proposed changes. $CO_2$ measurements paired with Bayesian inverse models have been shown to provide a quantitative assessment of emissions (Lauvaux et al., 2016; Lauvaux et al., 2020; Turner, et al., 2020a). To date, most attempts at quantifying urban $CO_2$ emissions have focused on extracting a temporally averaged (often a full year) total of the anthropogenic $CO_2$ across the full extent of city. A few studies have attempted to disaggregate emissions by sector or fuel type, or describe large shifts in aggregate emissions (Newman et al., 2016; Nathan et al., 2018; Lauvaux et al., 2020; Turner, et al., 2020a), but none characterize subsector processes of vehicle emissions.

High spatial density observations offer promise as a means to explore process-level emissions details. The BErkeley Air Quality and $CO_2$ Network (BEACO$_2$N) is an observing network deployed in the San Francisco Bay Area and other cities with measurement spacing of ~2km (Fig. 1, left). In a prior analysis, Turner et al. (2020a) showed that BEACO$_2$N measurements can detect variation in $CO_2$ emissions with time of day and day of week in addition to the dramatic changes in $CO_2$ emissions due to the COVID-related decrease in driving.

Here, we analyze hourly, spatially-allocated $CO_2$ emissions derived from the inversion of BEACO$_2$N observations (Turner et al., 2020a) to explore how well they constrain the $CO_2$ emissions from a 5km stretch of highway. This stretch chosen because of its location upwind of consistently active BEACO$_2$N sites and for completeness of traffic data, and because emission rates are highly affected by speed (vehicles use more fuel per km at very low and high speeds) and fleet-composition (HDV emit more $CO_2$ per km than light duty vehicles (LDV)). The variation of the ratio of total fleet $CO_2$ emission per vehicle km traveled (g $CO_2$ / vkm) is used to explore variations in on-road fuel efficiency and the factors responsible for that variation. We show that average fuel efficiency of the vehicle fleet on the road varies by as much as 27% over the course of a typical weekday.

## 2 Methods and Data

### 2.1 The Berkeley Air quality and CO₂ Network

We use hourly $CO_2$ observations from the Berkeley Air quality and $CO_2$ Network (BEACO₂N) (Shusterman et al., 2016; Kim et al., 2018; Delaria et al., 2021). The BEACO₂N network includes more than 70 locations in the SF Bay Area, spaced at ~2 km, and measures $CO_2$ with a network instrument error of 1.6 ppm or less (Delaria et al., 2021). All available data from January-June 2018-2020 are included in this analysis. During this time, more than 50 distinct locations had nodes that were active for a month or more (including 19 sites within 10 km of our highway stretch of interest). The number of nodes active

at any given time ranged from 7-41, with a mean of 17. Figure 1 shows sites in operation at some point during analysis period and Fig. S1 shows a timeseries of the number of nodes available throughout the study period.

### 2.2 The BEACO₂N - STILT Inversion System

To infer $CO_2$ emissions from within the BEACO₂N footprint, we use the Stochastic-Time Inverted Lagrangian Transport

(STILT) model, coupled with a Bayesian inversion as described in detail in Turner et al. (2020a). Briefly, we use meteorology from NOAA's HRRR product at 3 km resolution to calculate footprints from each hour at each site, weighted by a priori $CO_2$ emissions. The overall region of influence, the network footprint, as defined by a contour representing 40% of the $CO_2$ influence is shown in Fig. S2 (left). We construct a spatially gridded prior emissions inventory using point sources provided by the Bay Area Air Quality Management District (2011), home heating emissions as reported by

BAAQMD (2011) and distributed spatially according to population density, on-road emissions from the High-resolution Fuel Inventory for Vehicle Emissions (McDonald et al., 2014) varying by hour of week and scaled by year using fuel sales data, and a biogenic inventory derived using Solar Induced Fluorescence (SIF) Satellite data (Turner et al., 2020b).

To ensure a focus on highway emissions, we subtract prior estimates associated with non-highway sources from posterior BEACO₂N-STILT fluxes. Non-highway sources are small (~12%) in comparison with highway emissions for the

pixels corresponding with the highway stretch analyzed in this study (Fig. 2, left). We assume the error in prior estimates of these sources to be an even smaller fraction of the total. For reference, a diel cycle of sector-specific, weekday prior emissions for the pixels analyzed in this study is shown in Fig. S3.

We estimate the BEACO₂N-STILT inversion to be precise to at least 30% for a line source. This estimate is based on the results of Turner et al. (2016) who used Observation System Simulation Experiments to demonstrate that with 7 days

of observations at 30 sites a 45 tC/hr line source could be constrained to 15 t C/hr. However, this paper also demonstrated that error in the posterior decreased as results were averaged over a longer period of time. Here we are using 18 months, rather than 7 days of observations, we expect and observe better precision than 30%.

### 2.3 PeMS-EMFAC – derived CO₂ Emissions Estimates

Total hourly vehicle flow, truck (HDV) percent, and speed, were retrieved from http://pems.dot.ca.gov for January – June 2018-2020.  There are ~1800 traffic counting stations hosted by the Caltrans Performance Measurement System (PeMS) in the Bay Area, including more than 400 sites (Fig. S2) within the 2020 footprint of the BEACO2N, as described in Turner et al. (2020a). These stations count vehicle flow using magnetic loops imbedded in roadways and estimate HDV fraction using calculated vehicle speed and assumptions about vehicle length (Kwon et al., 2003). For hours during which

fewer than 50% of measurements were reported, we fill in total speed and light duty vehicle (LDV) flow gaps by using linear fits to nearest neighbor sites and gaps in HDV flow using hour-of-day- and weekend/weekday-specific median ratios between neighboring sites. We find that using this imputation method, mean absolute errors in speed are 5-10 km h$^{-1}$, in LDV flow are 500 vehicles h$^{-1}$, and in HDV flow are 50 vehicles / hour. (See Fig. S4.)

        We calculate both LDV and HDV vkm for each highway segment during each hour, using downloaded flow data at

each sensor location and segment lengths obtained from the PeMS database. For highway segments within the BEACO2N footprint, vkm are summed to obtain regional highway HDV and LDV vkm for every hour. Figure S2 (left) shows the extent of the PeMS network in comparison to the BEACO2N-STILT footprint, as well as total HDV vkm and LDV vkm.

        Vehicle fuel efficiency is dependent on both fleet composition and vehicle speed.  We calculate an emissions rate at each location by combining speed and the HDV percentage with fuel efficiency estimates provided by the California Air

Resources Board's Emissions FACtor Model (EMFAC2017). The EMFAC2017 model provides yearly fuel efficiency estimates for the Bay Area for 41 vehicle classes as a function of speed. We group these 41 vehicle types into the categories LDV or HDV. (Table S5) PeMS's vehicle-type classification system is length based, assuming that LDV have a median length of 3.7 m and HDV a median length of 18.3 m (Kwon et al., 2003). As a result, we group most light duty trucks into the LDV category. To find speed-dependent emissions rate values for the LDV and HDV groups, we find a vkm-weighted

mean of emissions rates across all vehicle-classes within a group at a given speed

$$er_{speed,group} = \frac{\sum_{i=1}^{n} vkm_{i,speed} er_{i,speed}}{\sum_{i=1}^{n} vkm_{i,speed}}, (1)$$

where $i$ is a vehicle class. From this, we generate LDV and HDV emissions rates at 8.02 km h$^{-1}$ (5 mph) intervals. (See Fig. S6.) EMFAC does not provide data for several LDV vehicle classes at and above 96.8 km h$^{-1}$ (60 mph). To fill in this gap, we estimate emissions rates for the LDV group by using emissions rate to speed slopes (g $CO_2$ vkm$^{-1}$ km h$^{-1}$ ) for high speeds

(88-145 km h$^{-1}$), using data from Davis et al. (2021).

        We calculate emissions rates (g $CO_2$ / vkm) for each (< 1km) road segment between PeMS sensors at a moment in time

$$er(t,seg) = \frac{vkm_{LDV}(t,seg)er_{LDV}(t,seg)+vkm_{tHDV}(t,seg)er_{tHDV}(t,seg)}{vkm_{LDV}(t,seg)+vkm_{HDV}(t,seg)}, (2)$$

where emissions rates for cars and trucks are found via spline fit between reported speed for that segment and time with our

curves for the emissions rates of each vehicle group. A fit is used rather than an individual bins, because of the sharp gradients that exist at low speeds for LDV. From the emissions rate for each (~1km) segment, we calculate an emissions rate for a stretch of highway including several segments to find total emissions rate (*er*) along a "stretch" over a period of time:

$$er(t, stretch) = \frac{\sum_{all\ segments}(vkm_{LDV}(t,s)er_{LDV}(t,s) + vkm_{HDV}(t,seg)er_{HDV}(t,s))}{\sum_{all\ segments}(vkm_{LDV}(t,s) + vkm_{HDV}(t,s))}. \quad (3)$$

Total $CO_2$ emissions rates for the highway stretch analyzed in this work are shown in Fig. 2 (right, bottom).

## 3 Results

To gain insight into the relative impacts of congestion and fleet composition, we calculate fleet-wide vehicle emission rates ($gCO_2$/vkm) using two different methods. For both methods, the Caltrans Performance Measurement System (PeMS) provides vehicle counts, speed and categorizes HDV vs. LDV (http://pems.dot.ca.gov). Using this data and estimates of fuel per km from the EMissions FACtor 2017 (EMFAC) Model, we calculate the $CO_2$ emissions per km for the average vehicle with hourly time resolution as described above. Second, we use the PeMS data in combination with g $CO_2$ per unit area derived from the BEACO$_2$N-STILT inversion system. We focus on the ~5 km stretch of Interstate-80 just north of the San Francisco-Oakland Bay Bridge (Fig. 2). Interstate 80 is an East-West Highway whose orientation in this stretch is mainly North-South, with eastbound lanes traveling north and westbound lanes traveling south. The road has 5 lanes in each direction and is often subject to high congestion (vehicles traveling slower than the posted speed).

PeMS-EMFAC-derived emissions rates give us insight into (1) the expected variation in emissions rates across a typical day (Fig. 2) and (2) the relative impacts of congestion vs. HDV percentage as factors leading to this variation (Fig. S7). For example, while the west-bound segment experiences speeds significantly below free-flow during both morning and evening rush hours, the east-bound segment experiences significant congestion only during the evening. Because of a steep gradient in LDV emission rates between 20 and 50 km h$^{-1}$ (Fig. S6), the west-bound congestion in this segment occurs at speeds that are more fuel efficient than free-flow. The overall variance in emissions rates over the whole stretch is significantly smaller than in either of the directions shown individually.

From PeMS-EMFAC-derived emissions factors, we predict a median diel cycle with emissions per km travelled ranging from ~247 to ~314 g $CO_2$ / vkm. For reference, if all vehicles were driving at the speed limit of 104.6 km h$^{-1}$ (65 mph) and the fleet mix was 6% HDV and 94% LDV, we calculate an emission rate of 265 g $CO_2$ / vkm. The range of predicted emissions are narrower on the weekend (238 to 276 g $CO_2$ / vkm), both because fewer HDV use the road and because there is a smaller range in speed.

Figure S7 shows the hourly variation in the relative contributions of LDV speed, HDV percentage, and HDV speed to the deviation in g $CO_2$ / vkm from the reference value of 265 g $CO_2$ / vkm. The solid line is the mean and the shaded envelope represents the day-to-day variance. In the morning and mid-day, HDV percentage and LDV speed have opposite impacts on g $CO_2$ / vkm, leading to small variations in g $CO_2$ / vkm over the day, despite substantial variations in the separate effects of speed and HDV %. During evening rush hour, low vehicle speeds result in higher emission rates, leading to large positive deviations. High day-to-day variance in vehicle speed contributes to high day-to-day variance in emission rates. At times near midnight, large, positive deviations are observed, mostly as a consequence of high HDV percentage, but

also because traffic flows at rates higher than 104.6 kph, leading to higher emission rates. Night-to-night variance in HDV
percentage is low, thus variance in nighttime predicted g $CO_2$ / vkm is small. HDV speed has little impact on g $CO_2$ / vkm.

We use $CO_2$ measurements from 50 BEACO$_2$N sites across the Bay Area, combined with the BEACO$_2$N-STILT inversion system to assess highway emissions from our stretch of interest. In Fig. 1, we show the location of BEACO$_2$N sites, the stretch of interest, and emissions estimates for this stretch. Note that the posterior emissions move substantially from prior emissions towards what is estimated from PeMS-EMFAC, particularly during evening rush hour, when the prior
overestimates emissions by ~20%.

We compare BEACO$_2$N-derived and PeMS-EMFAC-derived emissions rates ($CO_2$ / vkm) and find remarkable agreement. The PeMS-EMFAC-derived emissions rates range from 225-300 g $CO_2$ / vkm and include effects of both fleet composition and variation in speed. For BEACO$_2$N, we use the total $CO_2$ emissions from the inversion at times corresponding to narrow bins of PeMS-EMFAC g $CO_2$ / vkm. Figure 3 (left) shows an example of data selected at times with
with PeMS-EMFAC-derived fuel efficiency in the range 271.4-279 g $CO_2$ / vkm. There is a range of emissions at each vkm because of noise in the inversion, variation in speed and variation in fleet composition. The slope of a fit to the data in Fig. 3 (left) is an estimate of the emissions rate *(equation 4),* where *$CO_2$ emissions* is defined as hourly emissions summed over BEACO$_2$N pixels corresponding to our highway stretch of interest (Fig. 2)

$$er(\ g\ CO_2/vkm) = \frac{CO_2\ emissons}{vkm}. \quad (4)$$

Using 18 months of data for weekdays between 4 am and 10 pm, we compare PeMS-EMFAC-derived and BEACO$_2$N-derived $CO_2$ / vkm (Fig. 3, right). These hours were chosen, because they represent the hours for which we expect traffic emissions to be substantially larger than emissions from other sources in our area of interest (See Fig. S3). Fitting to a line forced through the origin, emissions rates found via the BEACO$_2$N inversion are within 3% (0.97 +/- 0.01) of those predicted using PeMS-EMFAC traffic counts. A more complete description of this fitting and error calculation
process can be found in Text S8 and a comparison to results from applying this method to the prior can be found in S9. Using the definition of limit of detection as three times our uncertainty, we calculate that we would be able to detect an 11% change in individual points (representing bins of fuel efficiency from a combination of HDV percent and speed) and a 3% change in the slope. Because 18 months of data was required to reach this level of certainty, if we assume the 2.3-3.8% per year decrease in emission rate found by Kim, et al. 2021, we should be able to detect a change in overall fuel efficiency with
three full years of BEACO$_2$N-STILT output.

We also consider how emissions rates compare throughout the day (Fig. 4, top). During the evening, PeMS-EMFAC-derived and BEACO$_2$N-derived emission rates are in good agreement. The BEACO$_2$N g $CO_2$/vkm increases from 256 g $CO_2$ / vkm before rush hour (2 pm) to 324 g $CO_2$/ vkm during peak rush hour (5 pm). Likewise, the PeMS-EMFAC-derived $CO_2$/vkm increases from 256 $CO_2$ / vkm to 320 $CO_2$ / vkm over the same time period. The BEACO$_2$N prior has a
slightly larger increase in emission rate over this period (256 g $CO_2$/vkm at 2PM to 361 g $CO_2$/vkm at 5PM). In contrast, during the morning rush hours, we see less agreement between PeMS-EMFAC-derived and BEACO$_2$N-derived emission

rate estimates. The BEACO2N inversion is similar to the PeMS-EMFAC estimate at 5 am local time (280 g $CO_2$ / vkm) and then the BEACO2N estimate increases over the morning rush hour to 330 g $CO_2$ / vkm at 8 am. This behavior is different than either the BEACO2N prior (175 at 5 am and 275 at 8 am) or the PeMS-EMFAC calculation which decreases over this

period (275 at 5 am and 250 at 8 am).

The discrepancy in the morning between emissions derived from PeMS-EMFAC and BEACO2N can potentially be reconciled by congestion. There is a non-linear relationship between vehicle speed and the rate of emissions. As such, congestion involving non-constant speeds can result in higher emissions than would be estimated using the average vehicle speed. This can be seen from a simple example. Consider two cases: 1) a LDV travelling at a constant 50 km h$^{-1}$ for one

hour and 2) a LDV traveling at 100 km h$^{-1}$ for 20 minutes and 25 km h$^{-1}$ for 40 minutes. Both vehicles travel 50 km in one hour and therefore have the same average speed. However, the emissions rate is 461.5 g $CO_2$/vkm at 25 km h$^{-1}$, 195 g $CO_2$/vkm at 50 km h$^{-1}$, and 221 g $CO_2$/vkm at 100 km h$^{-1}$. Using these emission rates, the vehicle in the first case would emit 9.75 kg $CO_2$ whereas the vehicle with the variable speed in the second case would emit 15 kg $CO_2$.

Contrasting the speeds (Fig. 4 bottom, right) during these two periods, we see that while both show a bi-modal

speed distribution, a greater fraction of morning speeds fall into the 40-100 kph range, whereas a greater fraction of evening speeds are < 40 km h$^{-1}$ or > 100 km h$^{-1}$. We show in Fig. S10, emission rate estimates based on hourly averaged speeds between 0-40 km h$^{-1}$ and 100-140 km h$^{-1}$ (more common in evening rush hour) are likely an upper bound on possible emission rates corresponding to those hourly averaged speeds, whereas emission rate estimates based on hourly averaged speeds between 40-100 km h$^{-1}$ (more common in morning rush hour) likely represent a lower bound of emissions. The

predicted range in emission rate resulting from non-constant speeds, combined with a larger HDV % in the morning (Fig. 4 bottom, right), is large enough to explain the mismatch observed during morning rush hour.

## 4 Discussion

Strategic reduction of emissions from transportation is important to both reducing total GHG emissions and improving AQ. To make informed decisions that reduce GHGs and exposure to poor AQ, policy makers need to know (1) how much is

being emitted, (2) location and timing of emissions, and (3) the relative impact of various sub-sector processes (vkm, fleet composition, congestion).

To effectively capture emissions from sub-sector processes, models are also reliant on emissions factor models, such as the EMFAC2017 emissions model used in this paper. While our BEACO2N-STILT based estimates largely agree with the EMFAC2017 emissions model for $CO_2$, tracking on-road changes in emission factors will be especially important as

the impacts of congestion and fleet composition evolve rapidly, making timely updates essential to creating spatially accurate inventories. For example, the EMFAC model predicts an 18% decrease in overall $CO_2$ emission rates by 2030, resulting from the improved fuel efficiency of combustion engine vehicles and a transition to hybrid and EV (~6.8% of LDV vkm and ~6% of HDV vkm are expected to be traveled by EV by 2030). While the increased share of hybrid and EV should

work to decrease the impact of congestion, a projected increase in total congestion and congested-vkm share by HDV (Texas A&M Transportation Institute, 2019) is likely to work against that trend, making the overall result difficult to predict.

To our knowledge, this paper represents the first demonstration that a high-density atmospheric observing network can both diagnose and quantify relative contributions of sub-sector processes at the neighborhood scale. We demonstrate that the BEACO$_2$N network (~2 km spacing) of low-cost $CO_2$ sensors, can be used to quantify emission rates at a specific location (~5 km stretch) and by time of day. We show that on the highway stretch, activity-based emissions estimates that account for speed and HDV % match the inference from atmospheric measurements to within 3%. Finally, we demonstrate that the BEACO$_2$N-STILT system detects daily changes in fuel efficiency that range from 200-300 g $CO_2$ / vkm and this system would be capable of detecting fleet-wide changes in fuel efficiency in ~3 years.

## 5 Outlook

In this work, we have demonstrated that the BEACO$_2$N-STILT system was able to infer emission rates from vehicles along a specific stretch of highway. To understand the extent to which this method can be applied to other contexts, future work should investigate the extent to which various elements of the BEACO$_2$N-STILT system, including measurement density, error in meteorology used to calculate STILT trajectories, and the quality of the prior, impact the ability of similar systems to estimate emissions.

For example, it is possible that the mismatch we observe during the morning rush hour may be due to a larger relative meteorological model error during the morning as compared to the afternoon and early evening in which the boundary layer is relatively well mixed. Because a highly mixed boundary layer is important for minimizing discrepancies between particle trajectories in the STILT model and real transport (Lin et al., 2003), inversions typically use only measurements taken during the afternoon, (Lauvaux et al., 2016; Nathan et al., 2019; Lauvaux et al., 2020) when the boundary layer is relatively well mixed. However, as discussed by Martin et al. (2019), the impacts of meteorological mismatch during the morning may be offset by stronger signal, and future work should explore the extent to which averaging results over long time periods or strategic filtering of meteorological mismatches can combat emissions error.

Beyond further exploration of the elements influencing the sensitivity and precision of the BEACO$_2$N-STILT system, because each BEACO$_2$N node measures CO, NO$_x$, and PM$_{2.5}$ in addition to $CO_2$ (Kim et al., 2018), the method presented in this paper has the potential to shed light subsector processes impacting emission factors of these co-emitted species. This is salient because plume-based emission factor measurements of co-emitted pollutants show various emissions factor models systematically underestimate emissions (Bishop, 2021), fail to capture spatial heterogeneity in these factors due to fleet composition (age and compliance with control technologies) for PM (Haugen et al., 2018; Park, et al., 2016) and Black Carbon (Preble et al., 2018), or fail to capture the impact of temperature on emissions factors.

Applying these methods across a broader spatial area and to other species (PM$_{2.5}$, NO$_x$, CO) should yield information of interest to both scientists and policy makers by:

1. Revealing spatial and temporal trends in emission rates and emission factors across an urban area and quantifying the contributions of congestion, fleet composition, or other factors to spatial variations.
2. Identifying and diagnosing the causes of traffic-related AQ hotspots that contribute to exposure inequities.
3. Tracking trends in the above over periods of years to decades.

**Author Contributions:**

HLF derived $CO_2$ emissions from traffic data, conceived of project design, wrote manuscript, collected $CO_2$ data. AJT created and ran $CO_2$ inversion code. HLF, JK, KC, ERD, CN, PW collected $CO_2$ data. RCC gave feedback on project design, assisted in writing manuscript.

**Competing Interest Statement:** We have no competing interests to disclose.

**Acknowledgments:** HLF was supported by NSF GRFP fellowship and Microsoft Research Internship. Thanks to K. Lauter and MSR Urban Innovation Group for support in thinking through PeMS data acquisition. AJT was supported as a Miller Fellow with the Miller Institute for Basic Research in Science at UC Berkeley. This research was funded by grants from the Koret Foundation and University of California, Berkeley. This research used the Savio computational cluster resource provided by the Berkeley Research Computing program at the University of California, Berkeley (supported by the UC Berkeley Chancellor, Vice Chancellor for Research, and Chief Information Officer). Thanks to HSK for reading through and offering organizational suggestions on the manuscript.

**Data Availability:** The $CO_2$ data used for this study are publicly available at http://beacon.berkeley.edu (Cohen Research, 2021). Raw data can be given upon request. The traffic data used for this study is publicly available at https://pems.dot.ca.gov/.

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

**Figures and Tables**

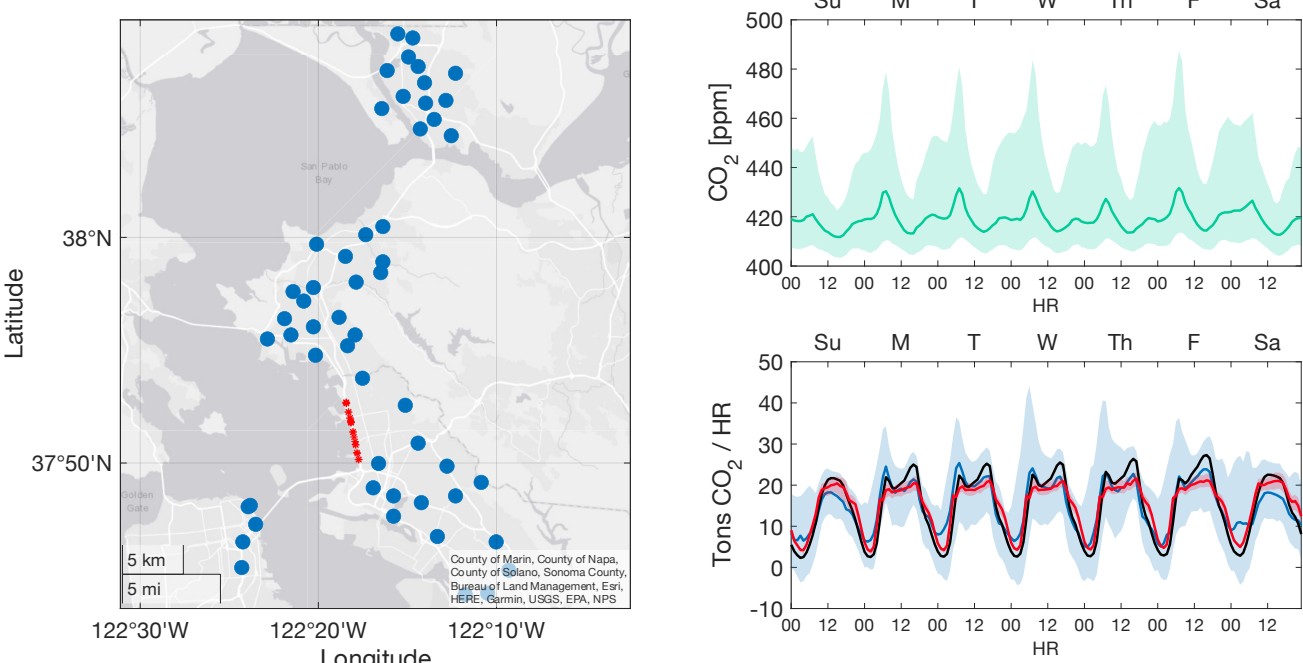

**Figure 1.** Left: Map of the BEACO2N Network shows all sites (blue dots) for which there are more than
4 weeks of data during the period analyzed (Jan-June 2018-2020). Red stars indicate location of PeMS
monitors used in this study. Right (top): $CO_2$ values shown for a 'typical week' during time period
observed. Dark line represents the median value observed across all sites and times. Shaded envelope
represents 1 sigma variance across the network and over the 2 year period. Right (bottom): $CO_2$
emissions on all highway pixels in the domain as derived from the inversion of BEACO2N observations
(blue), BEACO2N prior (black), and PeMS-EMFAC-based estimate (red). Shaded envelope shows
variance in emissions during the 18-month analysis window.

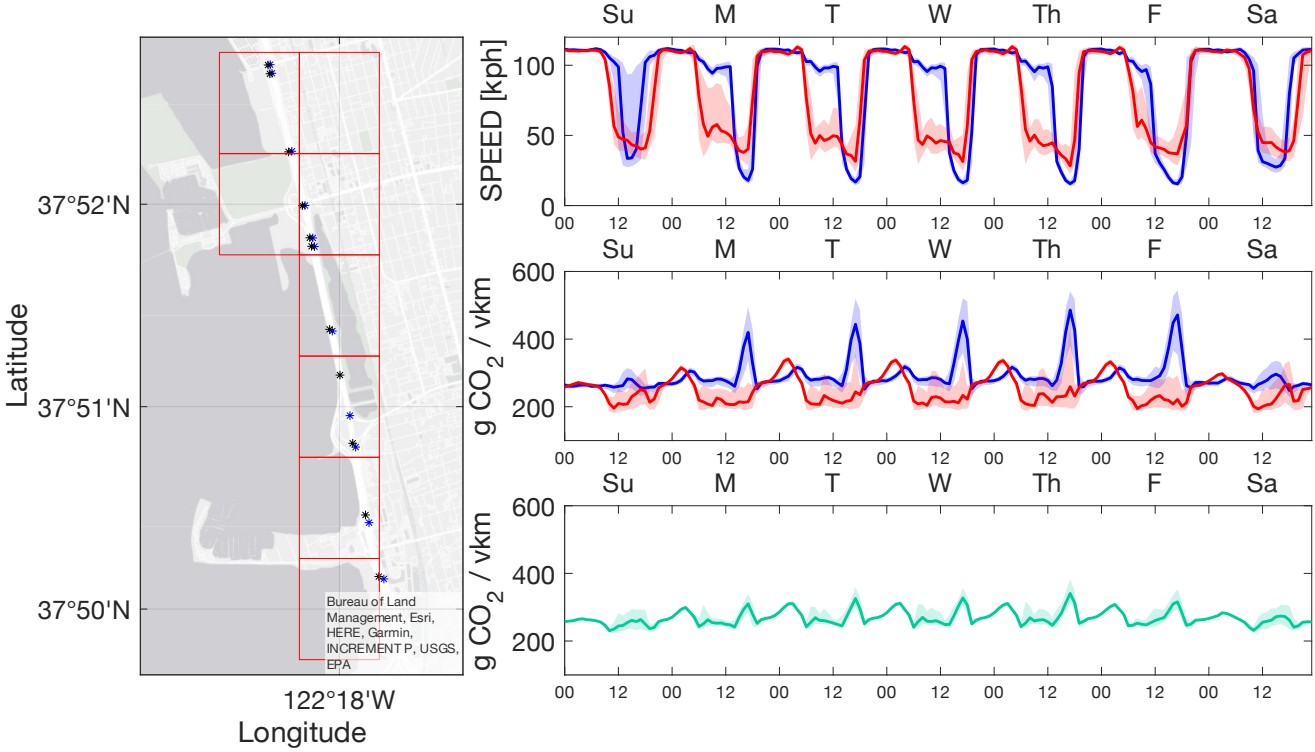

**Figure 2:** Left: ~5km stretch over which we analyze g $CO_2/vkm$. Points show the location of PeMS
stations. Squares show pixels associated with BEACO$_2$N STILT output which we use for comparison
for 5km stretch. Right (top): Hourly average speed shown for two opposite (West in red, East in blue)
PeMS measurement stations for a typical week. Right (middle): PeMS-EMFAC-derived emissions rates
calculated for two opposite (West in red, East in blue) PeMS measurement stations for a typical week.
Right (bottom): Aggregate PeMS-EMFAC-derived estimated emissions rates from the two directions of
traffic for a typical week for this highway stretch.

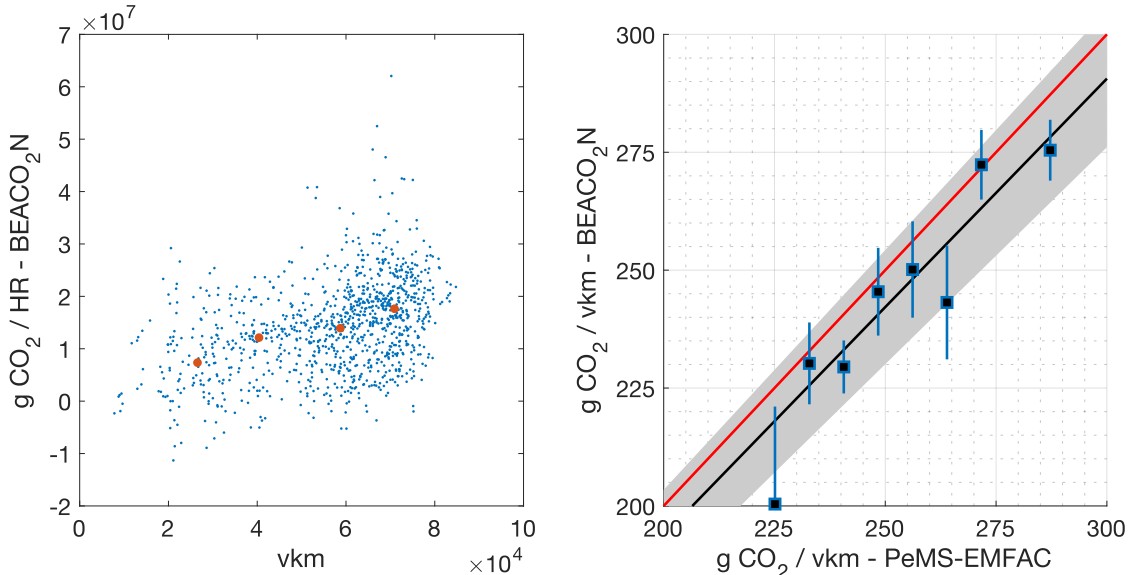

**Figure 3:** Left: BEACO$_2$N-derived emissions vs. vkm for times corresponding to modeled emission rates of 271.4-279 g CO$_2$/ vkm. Red points represent binned medians used in fitting. Right: BEACO$_2$N-derived vs. PeMS-EMFAC derived emissions rates with uncertainty estimate. Black line shows fit weighted by variance: y = 0.97(.01)x . Grey envelope is 5% deviation from fit. Red line represents 1:1 line.

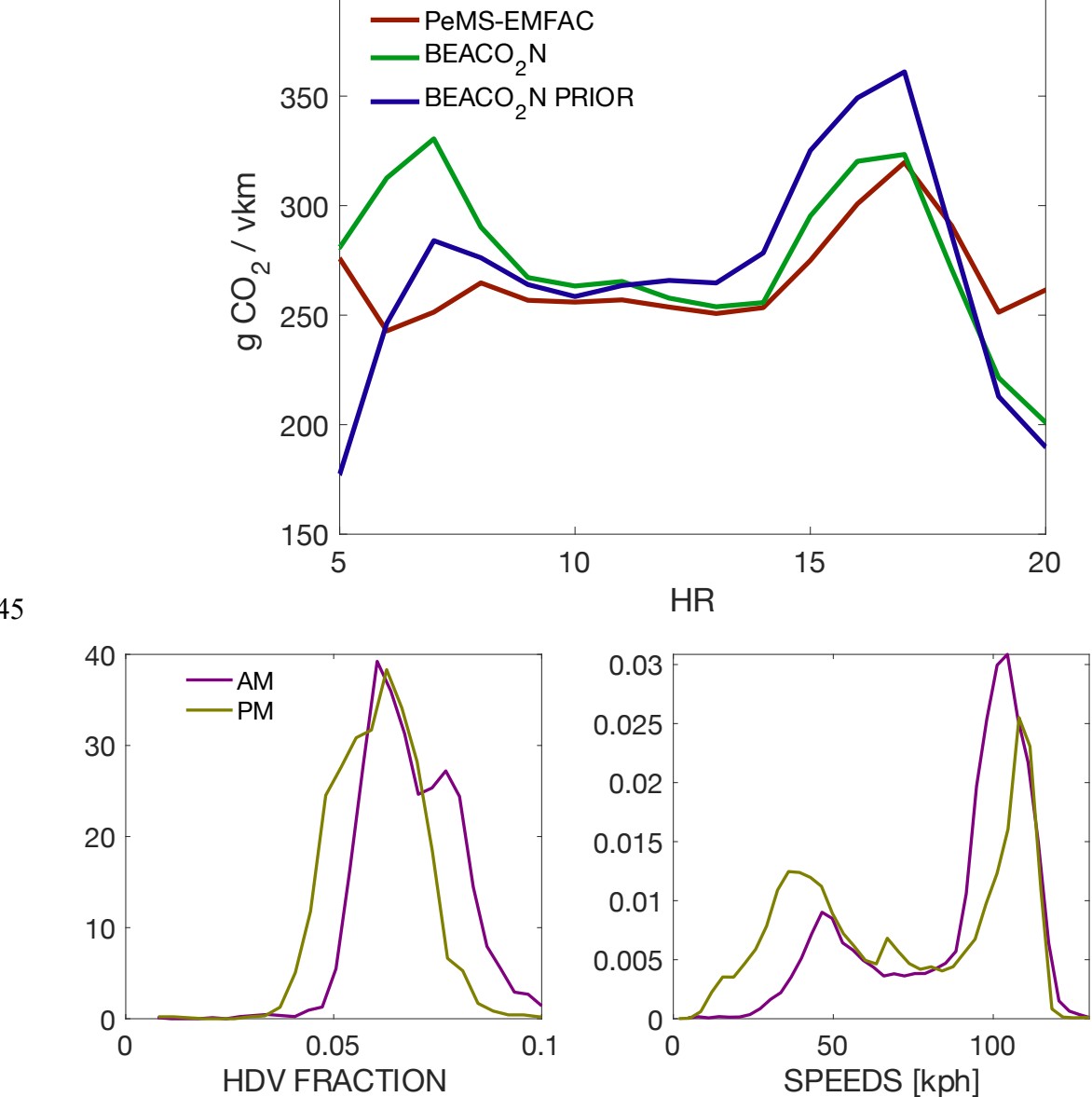


**Figure 4:** Top: Emissions rates by time of day on weekdays for PeMS-derived (red), BEACO2N-prior (blue), and BEACO₂N posterior (green). Bottom: Probability density functions of truck fraction (left) and speed (right) from weekday morning (5-9 am) and evening (4-8 pm) rush hour period on the
segment of I-80 analyzed in the Results section. Y-axis represents the relative probability of HDV fraction (left) or averaged hourly speed (right). Speeds are from individual PeMS sensors, while truck fraction is aggregated over the whole stretch under consideration (both directions).