# Peer review of "Assessing vehicle fuel efficiency using a dense network of CO2 observations"

_Atmospheric Chemistry and Physics, 2021_

## Author Comment (AC1)

We thank the referees for their comments and have included our response to the comments below. Original referee comments are in black and response is in red.

Response to Anonymous Referee #1

Referee comment on "Assessing vehicle fuel efficiency using a dense network of $CO_2$ observations" by Helen Fitzmaurice et al., Atmos. Chem. Phys. Discuss., https://doi.org/10.5194/acp-2021-808-RC1, 2021

Review of "Assessing vehicle fuel efficiency using a dense network of CO2 observations" by Fitzmaurice et al.

This manuscript describes a case study for which an inversion based on atmospheric measurements from a dense CO2 network was able to distinguish emissions characteristics over a strech of highway in the SF Bay area. Overall, the analysis is sound and promising. I had a few large-picture comments followed by specific comments, mostly pointing out some disorganization esp. on the SI.

We thank the reviewer for their constructive comments.

A few bigger-picture comments:

Overall, this is a very interesting outcome and example case of how atmospheric monitoring networks can achieve high-resolution understanding of emissions variability. I also appreciate the honesty in showing when the inversion matches the Pems analysis and when it does not, as well as the prior. Clearly, this work benefited from a good prior estimate of emissions that were adjusted using atmospheric data. I wondered what figure 3, right panel, would look like for the prior emissions estimates. In this context, did the inversion improve the prior estimate?

We have added Figure S9 to show what Figure 3, right looks like for the prior. For most circumstances the prior has constant fuel efficiency of 250g/km although there is some range built into the prior. When comparing to emission rate estimates created using the PeMS-EMFAC method described in the text, the posterior is a significant improvement over the prior.

Another larger comment: Could another reason for the early morning mis-match be larger meteorological model error early in the a.m. relative to the afternoon? Has this been investigated at all as a source of uncertainty in the inversion? Perhaps the authors could point to earlier work for this?

We agree that this is another possible reason for the mismatch. We believe that this is less likely. Over the 18 months of this analysis there is a wdie variation in morning PBL dynamics.

Looking at a larger area (40% contour of BEACO$_2$N footprint) over a shorter time period (Jan-June, 2020) we see that the greatest mismatch between BEACO$_2$N-STILT and PeMS occurs

during periods of stagnation (very low wind speed) observed at nearby weather stations but not represented by the modeled meteorology. These periods are not limited to the early morning. We are pursuing analyses that examine the accuracy of the observing/inverse modeling system and will report additional details in a forthcoming manuscript.

We have added references to and discussion of other work regarding the advantages of using data from the well-mixed afternoon in our outlook section. (L271-279)

Overall, I wonder if some caveats are required on the conclusions here, while not detracting from the very positive outcome overall. Here very good agreement is shown between methods, claiming that 5% emissions changes could be detected. Over what time frame could they be detected, and what does this level of capability depend on? i.e. a "good" transport/dispersion model, a dense network (this is already called out in line 239), a "good" prior - with "good" in quotations because likely how good a model or prior is needed is probably unclear. I think the paper could be improved by addressing some of these elements, or referring to past work that has shown the impact of things like the transport model or choice of prior, with a few sentences in the conclusion/discussion.

We add these caveats to the outlook section (L268-L271).

Specific Comments:

L83 might better reference https://doi.org/10.1029/2018JD029231, Nathan, B. J., Lauvaux, T., Turnbull, J. C.,
Richardson, S. J., Miles, N. L., & Gurney, K. R. (2018). Source sector attribution of CO2 emissions using an urban CO/CO2 Bayesian inversion system. Journal of Geophysical Research: Atmospheres

Agreed. We now cite this text.

Fig. 1 Left, can the authors add the meaning of the red dots to the caption?

The red stars represent PeMS measurement stations. This was added to the caption.

L104 Fig 1 does not seem to show the time series of the number of sites.? Reword, or add to the figure? It would be interesting to explain the impact of sites coming in and out during the inversion period, and how that might affect the inversion results in addition the average daily cycle of concentrations. I.e. if one site was particularly closer to a source than another, when the mix of sites changes, the average co2 would change as well.

We have re-worded and also added Figure S1 showing a time series of the number of sites. We agree that the relationship between site availability (and spatial cumulative influence at the location at which emissions are assessed) and inversion outcomes is an interesting question.

While it is beyond the scope of this paper to say much more, as mentioned above we are in the midst of an analysis that we hope to publish soon.

L119 over what area are non-highway sources 12% of total?

Non-highway sources represent 12% of total emissions over the pixels shown in Fig. 2 (left). We have added a note about this in the text.

L138 Figure 2 left doesn't actually show the extent of the PeMS network compared to the BEACON footprint. (the BEACON footprint I assume is much larger than this set of pixels). Clarify here what is shown. Or maybe this should refer to Fig S1, left?

Correct. This is described in the supplement. We have clarified the caption.

Fig S1 caption - I do not see a panel with CO2 emissions, only LDV and HDV?

We fixed this in the caption. CO2 emissions from PeMS are shown in Figure 1.

L146 missing a period.

Corrected.

L149 kph should probably be expressed as km h-1 (where -1 is a superscript)

Corrected.

L156, should this read "for the emissions rates for each vehicle GROUP"?, i.e. at this point the er from equation 1 is being used, which is the er for either LDV or HDV at this speed? Earlier these were defined as "groups", as they are a weighted average of er's from different classes within the group, right?

Corrected.

L156 regarding the spline fit, can the authors indicate why this was chosen? Earlier it was stated that the er's were calculated at 8 kph intervals, why is a piece-wise linear fit not sufficient for speeds between the intervals? (splines make me wary of over-fitting, what does this graph look like?).

The spline fit is very similar to the piecewise linear fit. We compare the two in Figure S6.

L157, how long are these segments and stretches, approximately

We added a text indicating segments are less than 1 km in length.

Eq. 3 some parentheses would be useful indicating whether both terms in the sum are being summed over all segments.

Parentheses added.

Fig 2, not being familiar with this roadway, it is not clear on the left which is East and which is West, as the highway seems to travel mainly North-South in this section? This could be clarified in the text around L167, (e.g. Interstate 80 is an East-West highway whose orientation in this stretch is actually mainly North-South, with the eastbound lanes traveling North etc etc). Or something clarifying as such.

Clarification added to text.

Fig 2 caption. Colon is missing after Right (bottom)

Colon added.

L174-180 this is interesting and makes the reader wish there were a figure showing the speed-dependent emissions profiles for LDV and HDV (also would answer L156 comment) - emission rates must increase substantially below some specific speed. Also, in Fig 2 it would be great to have a panel of the % LDV and HDV with time East vs. West to see what is driving the patterns in emissions shown here. This would address L171, where the authors note that the emission rates give insight into whether congestion or HDV percentage is the factor leading to variation - this is not shown in the figure at all, so there is no insight to the reader here. Otherwise, perhaps the authors could comment in the text on why the er of the East is so high at those low speeds in the evening, when the West is not that much faster but has very low er. (I do see the HDV and LDV time plots in Fig S1, but they don't show the difference between east and west nor showing just the fraction of HDV which sould be easier to show the effect here).

We add Fig S6 which shows emission rates as a function of speed for both LDV and HDV. There is a steep gradient in emissions rates for LDV from 20-30 kph (east) to 40-50 kph (west). This accounts for the east / west difference in Figure 2. HDV percentage accounts for a smaller overall fraction of emissions. (Fig S7)

We have added a sentence (L180) discussing these points.

Fig2 should the y-axes in the right lower panels be g $CO_2$ / km or vkm? I'm not sure really or maybe they are the same thing here.

We think of g $CO_2$ / vkm as being a property of the fleet and g $CO_2$ / km as being the property of individual vehicles, so vkm is appropriate here.

Fig 3, left, what are the red points? So the slope from the left figure is calculated and plotted in the right figure as one of the squares, and are the bars the error from the slope fit? What kind

of fit was used on the left figure slope (this is quite noisy, so could be sensitive to the fit - was it forced through zero, was it ODR or not?). [now having looked at the SI I see there is text addressing this, this should be referred to here, noting what the red points are].

We now reference the text in the SI addressing fitting details and have added description of red points to figure 3.

L195 why were those hours chosen?

These hours were chosen because vkm (and therefore $CO_2$ emissions) from vehicles are significantly higher for this period than for the hours between 11pm and 3am. Because of low vehicle emissions from 11pm – 3am, the ratio of vehicle emissions to other emissions is lower in the region of interest. As a result, any error in our prior estimate of non-traffic emissions would be aliased into a fractionally larger error in traffic emissions during these hours. See figure S3.

L200, mention this is for weekdays only (this is in the caption, but would be nice in the text as well).

We now mention this in L209.

L217 - One more sentence spelling this out would be useful, rather than letting the reader make a leap. The Pems method only uses hourly averages, so would have a lower emission rate than reality if reality were more like the second case? And here the authors are saying that it could be that morning rush hour exhibits this kind of large variability in speeds within an hour, leading to an incorrect estimate by the Pems method?

We further our description in L237-244.

Figure 4, lower panels don't seem to be addressed in the text anywhere. Figure S2 is not mentioned in the text either as far as I can tell.

We now address Fig 4 lower panels in L237 and L243.

Fig. S3 also is not mentioned but supports some of the text after L217 so should definitely be mentioned there! It also addresses an earlier comment I made asking for the plot of speed vs. emission rate.

We add a mention of the figure.

Text S3: Seems to describe a different figure? Fig. S3 does not show the dependence on HDV percentage - the caption says it is showing the results at 8% HDV. This figure would address some of my earlier comments.

Corrected.

The SI is a bit of a mish-mash, in that there is some text at the top that is not labeled with a header (S1 or whatever), and later there is a Text S3. Also Table S1 I do not think was cited in main text.

We have reorganized the SI so that sections follow the order in which they appear in the text and provided an introductory summary to the SI, describing each SI section and made sure that every section is referenced in the text.

---

## Author Comment (AC2)

We thank the referees for their comments and have included our response to the comments below. Original referee comments are in black and response is in red.

Response to Anonymous Referee #2

Referee comment on "Assessing vehicle fuel efficiency using a dense network of $CO_2$ observations" by Helen Fitzmaurice et al., Atmos. Chem. Phys. Discuss., https://doi.org/10.5194/acp-2021-808-RC2, 2021

General comments:

Fitzmaurice et al. al present a well-structured manuscript discussing the use of novel data collected in the atmospheric high-density network BEACO2N in combination with an inverse modelling framework to assess local highway CO2 emissions. They compare the CO2 emissions per vehicle kilometer travelled on the highway as derived from bottom-up modelling and their inverse framework. They find good agreement for many periods, but also noticeable deviations especially in periods with congestion. The daily changes in emissions are also tracked by both approaches and the study suggests that future emission trends due to local mitigation actions in the transport sector could be tracked.

The issue of mitigating greenhouse gas emissions is very timely and especially GHG emissions at urban scales have been moving into the focus of the atmospheric science community in recent years. This study is a nice addition and demonstrate novel capabilities to retrieve emission rate estimates at highly localized scale from a network of lower-cost sensors. Both quality and topic of the study are suitable for publication in ACP. However, there are quite a few minor technical issues and clarifications that need to be addressed before this manuscript should be accepted.

We thank the reviewer for these positive comments.

Air quality:

The introduction and discussion sections include paragraphs on the importance of air- quality proxies from traffic. However, the study itself only models (emission modelling) and estimates (inverse modelling) CO2 emissions. No AQ data is shown, so the discussion section feels speculative. Suggestion to either include some data on AQ/CO2 ratios collected during the period covered in this inversion or moving this into an outlook session to clarify that the AQ statements are not direct results from this work, but extrapolations.

We thank the reviewer for the suggestion of an outlook section and have moved most of the material to that section.

Non-highway sources:

The impact of non-highway sources needs to be addressed in a (little bit) more detail. The fact that is contributes ca. 12% on average is promising, but not sufficient.

Are those sources/sinks acting constantly? If they could vary on the short-term, i.e. with a very strong contribution during rush hours and no contribution any other time they could still influence e.g. the rush hour findings? (Unlikely but should be explicitly ruled out).

We have added a diel cycle of sector-specific prior emissions estimates for the area we consider (Fig. S3). The 12% value is integrated over the day. We expect other sources to make up a smaller fraction during rush hours, because traffic emissions are concentrated during those hours. We assume that point and area sources are constant over the day and that the biosphere changes by hour of day as described in Turner et al., 2020b.

Bibliography:

For some studies the author's first name are left our, sometimes they are abbreviated, sometimes even middle names are included. Citations should follow one consistent style, if possible.

Also, not all references fulfill the minimum criteria, e.g.: Boswell and Jacobson 2019: Where can this report be found?

Delaria et al. 2021 lacks information on where it was published. Please add journal name, issue, etc. or at least the DOI.

We have made the citations more consistent.

Minor and technical corrections:

L11: This statement is legitimate for the US, but it is unclear of transportation is also the largest CO2 source in developing economies, where urban centres also house large industrial and manufacturing districts.

We have clarified that we mean in the United States.

L32: (and throughout the manuscript): The citation style is very inconsistent. For some studies the author's first name are left our, sometimes they are abbreviated, sometimes even middle names are included. Citations should follow one consistent style, if possible. See general comments.

L38/L40: The authors state per capita emissions from vehicle have increased or stayed constant, but then cite a study that reports a 2% decrease. Please clarify if -2% is considered constant here. Or change to 'increased or stayed nearly constant'

Corrected to say "nearly constant."

L90: Why was a stretch of highway selected in the region with the lowest low-cost sensor density? Looking at figure 1 (left) most parts of the bay area have more sensors per km2.

We selected this highway stretch because (1) this site is upwind of many consistently active BEACO$_2$N sites, meaning that it is consistently in the BEACO2N-STILT inversion footprint throughout the time period studied, and (2) the PeMS data for this stretch was relatively complete compared to stretches of similar length in the region. Text was added to indicate this logic.

L101/102: the authors state that an accuracy of 1.6ppm was achieved, while Delaria et al. 2021 only reported: "a temperature-dependence correction, and a resulting network instrument error of 1.6 ppm CO2 or less". Accuracy seems less relevant than the network error, but if accuracy is reported, please clarify if further accuracy testing has occurred and if this was done against the latest WMO CO2 X2019 scale or an equivalent scale established by a National Metrological Institute.

We adjust our text to refer to network instrument error.

L119: Although non-highway sources are reported to be a minor contribution on average (ca. 12%) this is not sufficient information as it could maybe contribute a lot more during certain hours and a lot less during others. Please consider adding a diel cycle of the non- highway CO2 component to the main paper or report a range here, instead of the average of 12%.

We add this diel cycle to the SI. See SFig 3.

L133: Has this interpolation method been validated? It would be crucial to show that this linear interpolation works well or how much additional uncertainty it introduces. A quick test would be to choose a period with complete coverage, randomly remove 50% of data and see how we you can reproduce the true time series.

See added Figure S4.

L148 – equation 1.: This is unclear, shouldn't er_i be a speed dependant variable here? Or vehicle classes indeed vehicle-speed classes?

In the EMFAC2017 model, both er_i and vmt_i are given as speed dependent. We have changed subscripts to make this more clear.

L164: How well does the PeMS data reflect actual vehicle speeds on the highway? Is there a significant amount of uncertainty added here?

A larger amount of uncertainty comes from how representative hourly average speeds are of the speeds driven within the hour. We address this in Figure S10.

L167: See comment at L90

Addressed above.

L173: Please define congestion here. Is anything below free-flow considered congestion? For example, if the average speed on a segment is 60mph instead of the posted 65mph would that count as congestion?

Text added to define congestion as vehicle volume that results in lower than posted speed.

L194 – Equation 4: suggestion, do not $CO_2$ as variable to signify $CO_2$ emissions, as you already used it to signify $CO_2$ mixing rations (see Figure 1, right top). Maybe use E($CO_2$)/vkm =

We change to er (g $CO_2$ / vkm)

L197: Please elaborating the 5% assumption. 'Because eight of the nine points corresponding to emission rate bins fall within 5% of the fit, we estimate that the BEACO2N system would be able to detect a change in emissions rates of the order of 5%' Is there a statistical theorem that shows that this follows?

We remove this text and frame our statements around detection limit. (L216)

L203: please add 'g' to $CO_2$/vkm

Done.

L219: Air quality is not at all discussed in the results section of the manuscript and it is not included in the emission modelling or inversion results, so any discussion of it seems speculative – maybe better put in a 'outlook' section than a discussion. See general comment

Moved AQ discussion to an outlook section.

L225-229: Why are AQ proxies discussed here, although no data or modelling of AQ proxies is included in the study?

Moved AQ discussion to an outlook section.

L 243: see general comment on AQ

Moved AQ discussion to an outlook section.

L 247: see general comment on AQ

Moved AQ discussion to an outlook section.

L 267: change 'avail-able' to 'available'

Done.

L278/L285/L288/L325 please add required information for reference. E.g. DOI, website accessed, etc.

Done.

L353 – Figure 1: As the manuscript uses SI and SI-derived units I would assume that the label 'tons' in Fig. 1 right-bottom refers to 1000kg. If not please highlight the use of the common US short-ton.

All metric.

L360 – Figure 2: textbox in left figure nearly unreadable if printed on letter-size paper.

We have enlarged the textbox.

L368 – Figure 8: figure right shows only 8 points, while L 197 referred to 9 points. Why was one point omitted here?

That was a typo, now corrected. There are only 8 points total.